# Experimental evaluation of oxygen isotopic exchange between inclusion water and host calcite in speleothems

Ryu Uemura[1], Yudai Kina[2], Chuan-Chou Shen[3,4,5], Kanako Omine[2]

[1] Graduate School of Environmental Studies, Nagoya University, Furo-cho, Chikusa-ku, Nagoya, 464-8601, Japan
[2] Department of Chemistry, Biology, and Marine Science, Faculty of Science, University of the Ryukyus, 1 Senbaru, Nishihara, Okinawa, 903-0213, Japan
[3] High-Precision Mass Spectrometry and Environment Change Laboratory (HISPEC), Department of Geosciences, National Taiwan University, Taipei 10617, Taiwan ROC
[4] Research Center for Future Earth, National Taiwan University, Taipei 10617, Taiwan, ROC
[5] Global Change Research Center, National Taiwan University, Taipei 10617, Taiwan, ROC

*Correspondence to*: Ryu Uemura (ryu.uemura@nagoya-u.jp)

**Abstract.** The oxygen and hydrogen isotopic compositions of water in fluid inclusions in speleothems are important hydroclimate proxies because they provide information on the isotopic compositions of rainwater in the past. Moreover, because isotopic differences between fluid inclusion water and the host calcite provide information on the past isotopic fractionation factor, they are also useful for quantitative estimation of past temperature changes. The oxygen isotope ratio of inclusion water ($\delta^{18}O_{fi}$), however, may be affected by isotopic exchange between the water and the host carbonate. Thus, it is necessary to estimate the bias caused by this post-depositional effect for precise reconstruction of palaeo-temperatures. Here, we evaluate the isotopic exchange reaction between inclusion water and host calcite based on a laboratory experiment involving a natural stalagmite. Multiple stalagmite samples cut from the same depth interval were heated at 105°C in the laboratory for 0–80 hours. Then, the isotopic compositions of the inclusion water were measured. In the 105°C heating experiments, the $\delta^{18}O_{fi}$ values increased from the initial value by 0.7‰ and then remained stable after ca. 20 hours. The hydrogen isotope ratio of water showed no trend in response to the heating experiments, suggesting that the hydrogen isotopic composition of fluid inclusion water effectively reflects the composition of past dripwater. We then evaluated the process behind the observed isotopic variations using a partial equilibration model. The experimental results are best explained when we assumed that a thin $CaCO_3$ layer surrounding the inclusion reacted with the water. The amount of $CaCO_3$ that reacted with the water is equivalent to 2% of the water inclusions in molar terms. These results suggest that the magnitude of the isotopic exchange effect has a minor influence on palaeo-temperature estimates for the Quaternary climate reconstructions.

# 1 Introduction

Speleothems have provided invaluable terrestrial climate records over historical (e.g., (Zhang et al., 2008)) and glacial-interglacial (e.g., Wang (2001);Wang et al. (2017)) time scales. Among multiple climate proxies in speleothems (Fairchild et al., 2006), the stable oxygen isotope ratio of $CaCO_3$ ($\delta^{18}O_{ca}$) is the most widely used hydroclimate proxy. Changes in $\delta^{18}O_{ca}$ have been interpreted as proxies for monsoon strength (Wang, 2001), precipitation amount (Wang et al., 2017) or temperature (Dorale, 1998;Mori et al., 2018) depending on the hydroclimatic setting of each cave site.

This indeterminate nature of the $\delta^{18}O_{ca}$ interpretation is related to the fact that $\delta^{18}O_{ca}$ values are mainly controlled by two factors: the temperature in the cave and the annual mean $\delta^{18}O$ value of the rain water. To overcome this ambiguity, the isotopic compositions of water in fluid inclusions in stalagmites have been regarded as important proxies (Schwarcz et al., 1976;McDermott, 2004). Cave dripwaters are sealed in micro-scale cavities as fluid inclusions, whose $\delta^{18}O$ value is usually close to the $\delta^{18}O$ value of infiltration-weighted mean of local rain (Baker et al., 2019). Generally, stalagmites contain inclusion water, which accounts for 0.05 to 0.5 wt. % of a speleothem (McDermott, 2005). Thus, the isotopic compositions of fluid inclusions of stalagmites ($\delta^{18}O_{fi}$) trapped under humid conditions should directly preserve the $\delta^{18}O$ value of dripwater in the past.

$\delta^{18}O_{fi}$ values are also useful for estimating past temperature variations. Although experimental studies suggest that the $\delta^{18}O_{ca}$ values of speleothems are affected by kinetic effects (e.g., Hansen et al. (2019);Dietzel et al. (2009)), a modelling study suggests the temperature dependence of $\delta^{18}O_{ca}$ is within the range of −0.20 and −0.34 ‰/°C even under a kinetic condition (Mühlinghaus et al., 2009). The combination of the two isotopic compositions, $\delta^{18}O_{fi}$ and $\delta^{18}O_{ca}$, provides a direct estimate for the oxygen isotopic fractionation between calcite and water, which is controlled mainly by the formation temperature. Thus, variations in past temperature (i.e., not absolute values but relative changes) can be estimated on the assumption that the kinetic effect is constant during the speleothem formation period.

Early studies analysed the hydrogen isotope ratio of fluid inclusions water ($\delta D_{fi}$), and the $\delta^{18}O_{fi}$ value was inferred from the modern $\delta D$ vs $\delta^{18}O$ relation in meteoric water (Schwarcz et al., 1976;Harmon et al., 1979;Genty et al., 2002;Matthews et al., 2000;McGarry et al., 2004). These studies measured $\delta D_{fi}$ because of the technical difficulties associated with measuring the small amounts required to estimate $\delta^{18}O_{fi}$ values and a fundamental concern about the integrity of the $\delta^{18}O_{fi}$ data, namely, oxygen isotopic exchange between inclusion water and host calcite in speleothems.

After being trapped in the host calcite, inclusion water may continue to exchange oxygen isotopes with the surrounding host calcite as follows:

$$3H_2{}^{18}O + CaC^{16}O_3 \rightleftarrows 3H_2{}^{16}O + CaC^{18}O_3 \tag{1}$$

This isotopic re-equilibration effect potentially alters the $\delta^{18}O_{fi}$ values after the initial trapping of the inclusion water in the stalagmite. Such post-depositional isotopic exchange does not occur for hydrogen because of the small amount of hydrogen

in the host calcite. Many studies on geothermal water have shown positive $\delta^{18}O$ shifts from the global meteoric water line (GMWL) (Clark and Fritz (1999) and references therein). These data suggest an isotopic exchange of oxygen between the geothermal water and the host aquifer rocks. This observation is one of the reasons why the earlier studies mentioned above explored the $\delta D$ values of fluid inclusions rather than $\delta^{18}O_{fi}$ values (Harmon et al., 1979;Matthews et al., 2000;McGarry et al., 2004).

Recently, because of technological developments involving continuous-flow isotope ratio mass spectrometry (Vonhof et al., 2006) and cavity ringdown spectrometry (CRDS) (Uemura et al., 2016;Arienzo et al., 2013;Affolter et al., 2014), isotopic data from fluid inclusion have been accumulating. Interestingly, these studies suggest that post-depositional exchange is not significant. In fact, the fluid inclusions analyses of modern stalagmites have shown that their $\delta^{18}O_{fi}$ values are consistent with the $\delta^{18}O$ values of modern dripwaters (Griffiths et al., 2010;Dennis et al., 2001;Arienzo et al., 2013;Uemura et al., 2016;Labuhn et al., 2015), although a post-depositional alternation induced by recrystallisation has been suggested (Demény et al., 2016). Moreover, several studies successfully quantitatively estimated past temperatures at the time of calcite formation (van Breukelen et al., 2008;Meckler et al., 2015;Uemura et al., 2016). Therefore, these data imply that the oxygen isotopic exchange between inclusion water and stalagmite calcite appears to be limited and/or very slow over natural temperature ranges.

The indirect evidence for the insignificant post-depositional effect cannot provide explanations for the mechanisms behind this phenomenon. Thus, it is essential to investigate the isotopic exchange reaction and evaluate how much the past temperature reconstruction would be biased by this effect. However, to our knowledge, no study has investigated the magnitude of isotopic exchange within natural stalagmites. In addition to this natural process, the isotopic exchange reaction may occur during the drying process of stalagmite samples at high temperatures, which was commonly conducted before fluid inclusion measurement in the laboratory (Affolter et al., 2014;Vonhof et al., 2006;Uemura et al., 2016;Dublyansky and Spötl, 2009).

Here, we evaluate the isotopic exchange reaction between inclusion water and host calcite based on a laboratory heating experiment of natural stalagmite samples. We conducted a heating experiment because a higher temperature induces an increase in the degree of isotopic disequilibrium between water and calcite and increases the rate of isotopic exchange. As a result, an isotopic shift caused by the exchange reaction will be easier to detect. The stalagmite samples and experimental settings are described in Section 2. Section 3 presents the experimental results and discusses them using an isotopic exchange model for inclusion water and host calcite. Concluding remarks are given in Section 4.

## 2 Methods

## 2.1 Speleothem and dripwater samples

A stalagmite (named HSN1) was collected in Hoshino Cave, Minami-Daito Island, Okinawa, Japan (25° 51′ 34″N, 131° 13′ 29″E). The stalagmite is 246 mm in length, and most parts are milky white with thin transparent layers (Fig. 1). The

fabric of milky white layer is open columnar structure. To compare the isotope ratios of fluid inclusions after different heating times, ten layers, A-J, were taken (2.4-3.9 mm in thickness) from a quartered section of the stalagmite (Tables 1 and 2). The position relative to the axis may have an influence on the water content. Thus, to minimize this effect, 3-6 wedge-shaped chipped sub-samples (51-193 mg) were cut from each layer (illustrated in Fig. 1). The sub-samples from the same depth interval were assumed to have the same isotopic compositions.

Dripwater samples were collected in Hoshino cave from March 2011 to May 2016 (1-5 times per a year). At present, there is no dripwater at the sampling point of HSN1, the dripwater samples were collected at a point ca. 50 m away from the sampling point.

## 2.2 Isotope measurements

The oxygen and hydrogen isotopic compositions of the water in the fluid inclusions were measured using a homemade extraction device (Uemura et al., 2016). Briefly, the speleothem sample was crushed under vacuum, and then extracted water vapour was transferred to a CRDS isotope ratio analyser (L2130-I, Picarro Inc.) at the University of the Ryukyus. Compared with the system described in (Uemura et al., 2016), the system has been improved in three ways. (1) The valves were automatically controlled using pneumatic valves. (2) The entire system was heated at 105°C because most parts were easily damaged at 150°C. (3) The water vapour extracted from the stalagmite was immediately trapped cryogenically at the temperature of liquid nitrogen, thereby preventing interaction between $CaCO_3$ and water vapour. The $1\sigma$ reproducibility for the inclusion water analysis, based on the replicate analyses, was ±0.3‰ for $\delta^{18}O$ and ±1.6‰ for $\delta D$ (Uemura et al., 2016).

The $\delta^{18}O$ and $\delta D$ values were measured simultaneously using a cavity ring-down spectrometer (L2130-i, Picarro Inc.) with a vaporizer unit (V1120-i, Picarro Inc.). The $1\sigma$ reproducibility, based on repeated analyses of a working water standard water, was ±0.08‰ for $\delta^{18}O$ and ±0.26‰ for $\delta D$.

Stalagmite carbonate sub-samples were measured using an isotope ratio mass spectrometer (Thermo DELTA V Advantage) equipped with a Thermo Gasbench II system at the University of the Ryukyus. Powdered sub-samples of 150-200 μg were reacted with 100% phosphoric acid at 72°C in septa-capped vials before measuring the released $CO_2$. The $1\sigma$ (n = 124) reproducibility for the analysis was ±0.04‰ for $\delta^{18}O$ and ±0.03‰ for $\delta^{13}C$ based on repeated analyses of a carbonate standard (IAEA CO-1).

The isotopic composition is expressed in units of per mill (‰) using delta notation ($\delta = R_{sample}/R_{ref} - 1$), where R is the isotopic ratio and $R_{sample}$ and $R_{ref}$ are the isotopic ratios of the sample and reference, respectively. For water samples from fluid inclusions, $\delta^{18}O$ and $\delta D$ data are presented relative to Vienna standard mean ocean water (VSMOW) and are normalized to the VSMOW-SLAP scale. For calcite samples, $\delta^{18}O$ and $\delta^{13}C$ data are expressed relative to the Vienna Pee Dee Belemnite (VPDB) and/or VSMOW references. For clarity, the $\delta$ values of fluid inclusion water and stalagmite calcite are expressed with the subscripts "fi" and "ca", respectively. $\delta^{18}O_{fi}$, for example, indicates the $\delta^{18}O$ value of the water in fluid inclusions.

Two layers (depths of 75.0 mm and 190.0 mm) of the HSN1 stalagmites were dated using U-Th techniques (Shen et al., 2008;Shen et al., 2012). U-Th isotopic compositions and concentrations were measured on a multi-collection inductively coupled plasma mass spectrometer (NEPTUNE, Thermo-Fisher Scientific Inc.) at the High-Precision Mass Spectrometry and Environment Change Laboratory (HISPEC) of the Department of Geosciences at National Taiwan University (Shen et al., 2012). U-Th ages were calculated based on decay constants, half-lives, and $^{238}U/^{235}U$ ratios (Jaffey et al., 1971;Cheng et al., 2013;Hiess et al., 2012). Uncertainties in the U-Th isotopic data and $^{230}Th$ dates (yr BP, before 1950 AD) are given at the two-sigma ($2\sigma$) level or two standard deviations of the mean ($2\sigma_m$) unless otherwise noted.

## 2.3 Heating experiment

To evaluate the effect of isotopic exchange between water and host calcite, a heating experiment was conducted. As a pretreatment process, each stalagmite sample was placed in a small glass tube (o.d. 1/2 inch and 6 cm in length) with an airtight screw cap sealed with two Viton O-rings. Then, the samples were dried in a vacuum line for 17 hours at room temperature using a turbo molecular pump (with a pressure down to $10^{-5}$ Pa).

After this pre-drying process for all samples, the sample tube was subjected to different temperatures of 105°C or 25°C. For the 105°C heating experiment, the sample tube was heated with a silicone cable heater at 105°C for a defined time period (from 0 to 70 hours) under the same vacuum conditions (i.e., a pressure down to $10^{-5}$ Pa). For comparison, we also performed a non-heating experiment at room temperature (25°C) for the same time periods (from 0 to 70 hours) under the same vacuum conditions.

Then, the heated (or non-heated) sample was transferred in the closed sample tube to the fluid inclusion analysis apparatus so that atmospheric exposure when introducing the sample lasted for less than 30 seconds. To evacuate the analytical line, the samples were subjected to an additional 105°C condition for 20 minutes. Then, the sample was crushed, and the isotopic composition of the inclusion water was measured as described in Section 2.2.

## 3 Results and discussion

## 3.1 Isotope composition of inclusion water

The temporal variations in $\delta^{18}O_{fi}$ and $\delta D_{fi}$ values resulting from the 105°C heating and non-heating experiments are shown in Fig. 2. The isotopic compositions of inclusion water are shown as a deviation from the initial value ($\Delta\delta^{18}O_{fi}$ and $\Delta\delta D_{fi}$). For the 105°C heating experiment, the $\delta^{18}O_{fi}$ values gradually increased with the heating time and then reached a constant value after ca. 20 hours (Fig. 2a). The regression curve of the data represents an exponential function (Section 3.5). The $\delta^{18}O_{fi}$ values increased by ca. 0.7‰ with respect to the initial values during ca. 20 hours from the start of heating (Fig. 2a). In contrast, no significant trend was found for $\Delta\delta D_{fi}$ for the 105°C heating experiment (Fig. 3b). For the room-temperature drying experiment, neither $\Delta\delta^{18}O_{fi}$ nor $\Delta\delta D_{fi}$ values showed any systematic variation (Fig. 2c and 2d).

Overall, the results suggest that the observed increase in $\delta^{18}O_{fi}$ values in the 105°C heating experiment is caused by the oxygen isotopic exchange between inclusion water and the surrounding calcite. The data from the control experiment at room temperature suggest that the oxygen isotopic exchange reaction is too small to detect under 25°C conditions. The $\delta D_{fi}$ data for the 25°C experiment confirms that there is no systematic isotopic variation caused by subsample cutting and length of drying time. In addition, the $\Delta\delta D_{fi}$ value for the 105°C experiment did not change because there is no significant hydrogen reservoir in calcite.

The new calcite precipitation in fluid inclusions did not occur because the $\delta^{18}O_{fi}$ is expected to be lower if the new calcite, whose $\delta^{18}O$ value is higher than that of water, formed inside the inclusions. This is opposite to the result of heating experiment. In the case of internal calcite dissolution, the $\delta^{18}O$ value of water, will be changed through the isotopic exchange reaction between the bicarbonate in the solution and the water. Thus, if the amount of dissolution is limited, it is not different from the case that the water is re-equilibrated with a limited amount of $CaCO_3$ (will be discussed in section 3.4).

### 3.2 Evaporation during heating

Long-term heating may induce leakage from water inclusions through microscopic channels in the calcite caused by decrepitation of the calcite. The measured water content (weight ratio of water in fluid inclusions to carbonate) of the experiments is shown in Fig. 3. Overall, the water content ranged from 0.05 to 0.3 wt. %, which is within the typical observed range of 0.05-0.5% (McDermott, 2005).

Although there are large variations in water content among different layers, there is no significant trend between heating time and water content (Fig. 3). This result suggests that the fluid inclusion water does not evaporate/leak even after long-term heating. We should note that larger data sets of various stalagmites are needed to generalize this result because the behaviour of leakage would also be influenced by the fabric and micro-structure of the stalagmite. In addition, this finding suggests that our standard drying procedure (17 hours at room temperature, as described in Methods) is enough to remove the water adsorbed onto the calcite. Therefore, this result confirms that the increase in $\delta^{18}O_{fi}$ values (Fig. 3) in the heating experiments is not caused by evaporation due to thermal decrepitation.

### 3.3 Evidence of a post-depositional effect in the D-O plot

The distinct behaviour of $\delta D_{fi}$ and $\delta^{18}O_{fi}$ values is clearly depicted in the $\delta D$-$\delta^{18}O$ plot (Fig. 4). The GMWL, local meteoric water line (LMWL), and the precipitation-weighted annual mean values (2009-2012) of the rain water on Okinawa-jima Island are shown in Fig. 4. The LMWLs for summer and winter seasons are calculated based on the rain water data from Okinawa-jima Island (Uemura et al., 2012), ca. 400 km west of Minami-daito Island. The present-day dripwater isotopic data of Hoshino cave are close to the higher ones of annual mean precipitation values on Okinawa-jima (Fig. 4). Generally, the initial $\delta D_{fi}$ and $\delta^{18}O_{fi}$ values in the 25°C experiment are scattered between the summer and winter LWMLs.

The Thorium-230 dating results of stalagmite HSN1 from Hoshino Cave are shown in Table 3. The ages of layers at depths of 75.0 and 190.0 mm were 6429±55 and 7092± 48 years BP, respectively. Thus, a more detailed comparison

between the HSN1 fluid inclusions and the present-day rainwater is not straightforward because the HSN1 speleothem was grown during mid-Holocene and the rainwater isotope ratio is likely different from that of modern rainfall.

The important characteristic of this result is that the $\delta^{18}O_{fi}$ values of the heating experiment are systematically shifted in the isotopically enriched direction. The average $\delta^{18}O_{fi}$ value of the heating experiments ($-5.4\pm0.4‰$) is higher than that of the 25°C experiment ($-6.1\pm0.3‰$) by 0.7‰. In contrast, the average value of $\delta D$ does not differ between these experiments ($-34.5\pm1.9‰$ in the heating experiment, $-33.9\pm1.3‰$ in the room-temperature experiment).

Such positive $\delta^{18}O$ shift is opposite to the negative $\delta^{18}O$ shift of inclusion water in speleothems possibly induced by recrystallisation (Demény et al., 2016). Instead, the positive shift is similar to the $\delta^{18}O$ shift from GMWL found in observations of geothermal water (Clark and Fritz, 1999). Although the magnitude of the shift is much larger for geothermal water, 5-15‰, the $\delta^{18}O_{fi}$ shift found in our experiment is likely caused by the exchange of $^{18}O$ between the inclusion water and the host calcite. The possible reasons for the small magnitude of the $\delta^{18}O$ shift, 0.7‰, in our experiment will be discussed later (Section 3.4 and 3.5).

In fluid inclusion studies, closeness to the LMWL in $\delta D$-$\delta^{18}O$ plots has been used as proof for the validity of analytical methods and the integrity of the sample. Our result calls for caution regarding the $\delta D$-$\delta^{18}O$ plot test. Most of the inclusion data from the heating and room-temperature experiments are distributed between the summer and winter LWMLs. As discussed above, the artificial increase in the $\delta^{18}O$ value by heating is systematic. However, it is difficult to detect such a shift in the $\delta D$-$\delta^{18}O$ plot because the shift is small in the scattered data points. The data from the heating experiment (more than 10 hours) plot outside the summer LMWL, but a 0.7‰ deviation could be interpreted as resulting from different climate conditions. Because the oxygen isotopic exchange results in higher $\delta^{18}O$ values without a $\delta D$ shift, the isotopic exchange results in a lower deuterium excess value ($d = \delta D - 8\ \delta^{18}O$), $-6‰$ shift in our experiment. We should note that the $d$ value could become higher if the exchange takes place at lower temperatures than the original precipitation temperature. Therefore, the oxygen isotope exchange under changing temperatures may cause any slight deviation from the LMWL.

**3.4 Partial isotopic exchange between water and calcite**

To interpret the experimental result, we consider two hypotheses: (i) the oxygen isotopic exchange reaction occurred between inclusion water and the entirety of the host calcite (hereafter referred to as the "fully reacted hypothesis") and (ii) the $\delta^{18}O_{fi}$ values are equilibrated with a limited amount of $CaCO_3$ (hereafter referred to as the "partially reacted hypothesis"). These hypotheses are schematically explained in Fig. 5.

For the fully reacted hypothesis, the number of oxygen atoms in calcite can be considered infinite compared with that in the water inclusions because the water content of a stalagmite is very small: 0.05-0.5 wt. % (McDermott, 2005). Thus, the $\delta^{18}O_{fi}$ is simply controlled by the $\delta^{18}O$ value of calcite and the equilibrium fractionation factor between $CaCO_3$ and $H_2O$ at the ambient temperature (Fig. 5b). At 105°C, the fractionation factor between $CaCO_3$ and water is 1.0167. Because the $\delta^{18}O_{ca}$ value of HSN1 stalagmite is 25.5‰ vs VSMOW (i.e., $-5.3‰$ vs VPDB), the $\delta^{18}O_{fi}$ value in equilibrium with the calcite should be 8.6‰ vs VSMOW. Thus, $\delta^{18}O_{fi}$ value should be enriched by 14 to 15‰. This simple hypothesis, however, is not

realistic because the inclusion water likely reacts only with the inner surface of $CaCO_3$ surrounding the inclusions. In fact, the results in Fig. 2a show that the actual change in the $\delta^{18}O_{fi}$ value is only 0.7‰.

The small $\delta^{18}O_{fi}$ shift observed in the experiment can be interpreted as the result of (i) insufficient reaction time between water and calcite and/or (ii) reaction with a limited amount of $CaCO_3$. The first hypothesis can be rejected because the $\Delta\delta^{18}O_{fi}$ enrichment plateaued within 20 hours (Fig. 2a). Therefore, the experimental results support the latter hypothesis.

A model for this partially reacted hypothesis will be discussed in the following section.

### 3.5 Isotopic re-equilibration model

We describe a partial isotopic equilibration model that considers the changes in $\delta^{18}O_{fi}$ values from the time when the inclusion water was entrapped in $CaCO_3$ (time 0) to a certain time required to reach a new isotopic equilibrium (time $t_1$; at a certain temperature $T_1$). An example of this re-equilibration scenario is that inclusion waters at room temperature (time = 0,

temperature = $T_0$) are heated to 105°C ($T_1$) and reach a new isotope equilibrium at time $t_1$.

For the partially reacted hypothesis, we assumed that a limited amount of $CaCO_3$ in the reacted layer exchanged oxygen isotopes with the inclusion water (Fig. 5a). The isotope mass balance between the initial and partially equilibrated conditions at a heating time during a course of reaction at time t can be written as follows:

$$3 \cdot \gamma \, \delta^{18}_{ca\_ra(0)} + \delta^{18}_{fi(0)} = 3 \cdot \gamma \, \delta^{18}_{ca\_ra(t)} + \delta^{18}_{fi(t)} \tag{2}$$

where $\gamma$ is a molar ratio of $CaCO_3$ involved in the reaction to the inclusion water (i.e., $\gamma$ = $CaCO_3$ in reacted layer / $H_2O$ in inclusion: $M_{CaCO3}/M_{H2O}$), $\delta^{18}_{ca\_ra(t)}$ indicates the $\delta^{18}O$ value of the $CaCO_3$ in the reacted layer, and $\delta^{18}_{fi(t)}$ indicates the $\delta^{18}O$ value of the inclusion water at time $t$. If the $CaCO_3$ does not react with the water (i.e., $\gamma$ = 0), for example, the $\delta^{18}O_{fi}$ value

does not change before and after the reaction.

At $t_1$, the $\delta^{18}_{ca\_ra(t)}$ and $\delta^{18}_{fi(t)}$ reach an isotopic equilibrium state. This equilibrium state can be written as:

$$\delta^{18}_{ca\_ra(t1)} - \delta^{18}_{fi(t1)} = \varepsilon^{18}_{T1} \tag{3}$$

where $\varepsilon^{18}_{T1}$ indicates the oxygen isotopic enrichment factor at the temperature $T_1$. Whereas the rate constant of the isotope exchange reaction only varies with temperature, the number of transferred isotopes varies with the temporal evolution of the isotope ratios of the end members. The reaction can be written as:

$$-\frac{d(\delta^{18}_{ca\_ra(t)} - \delta^{18}_{fi(t)} - \varepsilon^{18}_{T1})}{dt} = k_{18}(\delta^{18}_{ca\_ra(t)} - \delta^{18}_{fi(t)} - \varepsilon^{18}_{T1}) \tag{4}$$

where $(\delta^{18}_{ca\_ra(t)} - \delta^{18}_{fi(t)})$ is the isotopic difference, $\varepsilon^{18}_{T1}$ is the isotopic enrichment factor at the new equilibrium state, and $k_{18}$ indicates a reaction constant. The integration of Eq. (4) yields:

$$\left(\delta^{18}_{ca\_ra(t)} - \delta^{18}_{fi(t)} - \varepsilon^{18}_{T}\right) = \left(\varepsilon^{18}_{T0} - \varepsilon^{18}_{T1}\right) \exp\left(-k_{18}t\right) \tag{5}$$

where $\varepsilon^{18}_{T0} (= \delta^{18}_{ca\_eq(0)} - \delta^{18}_{fi(0)})$ represents the oxygen isotopic enrichment factor at the initial temperature of $T_0$.

To fit our experimental data, in which we measured the difference between heated and initial (i.e., non-heated) conditions, we took the difference between Eq. (5) and the initial condition ($t = 0$) of Eq. (5) ($\delta^{18}_{ca_{eq(0)}} - \delta^{18}_{fi(0)} - \varepsilon^{18}_{T1} = \varepsilon^{18}_{T0} - \varepsilon^{18}_{T1}$). Thus, with Eq. (2), we obtain

$$\Delta\delta^{18}_{fi(t)} = \left(\frac{3\gamma}{1+3\gamma}\right)\left(\varepsilon^{18}_{T0} - \varepsilon^{18}_{T1}\right)\left(1 - \exp(-k_{18}t)\right) \tag{6}$$

where $\Delta\delta^{18}_{fi(t)}(= \delta^{18}_{fi(t)} - \delta^{18}_{fi(0)})$ represents the $\delta^{18}O_{fi}$ difference between the reacted and initial conditions. When the new equilibrium state is reached ($t = t_1$), Eq. (6) yields:

$$\Delta\delta^{18}_{fi(t1)} = \left(\frac{3\gamma}{1+3\gamma}\right)\left(\varepsilon^{18}_{T0} - \varepsilon^{18}_{T1}\right) \tag{7}$$

This equation provides an estimate of $\gamma$ based on the experimental results.

A regression curve based on Eq. (5), is shown in Fig. 2a. Based on Eq. (7), the value of $\gamma$ is estimated to be 0.02. This suggests that the amount of $CaCO_3$ reacted with water is equivalent to 2% of the water inclusions in molar terms. The thickness of the reacted layer of $CaCO_3$ can be roughly estimated on the assumption that the fluid inclusions filled with water are cubic with 50 μm edges. With a calcite density of 2.71 g/cm$^3$, the thickness of the reacted layer of $CaCO_3$ is estimated to be 0.6 μm.

This result is explained schematically in Fig. 5a and 5c. The $\delta^{18}O_{fi}$ value increased by only 0.7‰. Since the equilibrium fractionation factor between $CaCO_3$ and water at 105°C is 1.0167, the $\delta^{18}O_{ca}$ of the reacted layer should be 11.3‰ vs VSMOW. Thus, in this case, the $\delta^{18}O_{ca}$ value of the reacted layer changed significantly.

## 3.6 Impact for palaeo-climate reconstruction

In this section, we estimate the impact of the isotopic exchange effect on Quaternary palaeo-climate reconstructions. First, at 105°C, detectable isotopic exchange occurred within 20 hours. This finding suggests that researchers should be aware of this effect during experimental procedures, such as the heat drying process, before fluid inclusion measurements. Second, we consider a case in which the dripwater enclosed in the fluid inclusions during a glacial period with a temperature

of 15°C re-equilibrates at the modern average temperature of 25°C. The isotopic enrichment factors ($\varepsilon_T^{18}$) for 15°C and 25°C are 30.6‰ and 28.4‰, respectively. Thus, based on Eq. (7) with a γ value of 0.02, the fluid inclusion oxygen isotope ratio $\delta^{18}O_{fi}$ would increase by only 0.1‰. With the temperature dependence of the $\delta^{18}O$ fractionation factor between water and calcite (0.2‰/°C), this isotopic exchange results in a 0.5°C bias in the palaeo-temperature estimate. We should note that we do not know the reaction rate of the isotopic exchange under normal ambient temperatures. Calcite and fluid inclusion water might not reach a new equilibrium state even after thousands of years. Thus, this estimate is an upper limit of the bias. Because the bias, 0.1‰, is within the typical analytical error for inclusion analyses, the post-depositional effect has little influence on the palaeo-temperature estimates for glacial-interglacial cycles.

## 4 Conclusions

Our experiment shows that the $\delta^{18}O$ values of fluid inclusion water in speleothems changes because of isotopic exchange reactions with the host calcite. Unlike the $\delta^{18}O$ value of inclusion water, the δD value showed no trend even after prolonged heating and thus effectively reflects the original isotopic composition of past dripwater. This study is the first to present experimental data showing that such a post-depositional effect occurs in natural speleothem samples. However, the changes in the $\delta^{18}O$ values of fluid inclusion were very small, 0.7‰, in the 105°C heating experiment. Based on this result, the inclusion water reacts only with a thin layer of surrounding $CaCO_3$. The amount of $CaCO_3$ that reacted with the water is equivalent to 2% of the water inclusions in molar terms. Thus, the oxygen isotopic exchange results in a minor impact on the estimation of past temperature changes: a maximum bias of 0.5°C for a 10°C climate shift. This study provides a quantitative explanation of the mechanism by which the effect of isotopic exchange appears to be insignificant in previous speleothem studies. The results also suggest that the sample treatment in the laboratory (i.e., heated drying process) should be conducted with caution because isotopic exchange may affect the $\delta^{18}O$ value of fluid inclusions.

**Data availability**

The data generated and used in this study are available in Tables 1 and 2 in this article.

**Author contributions.**

RU designed the experiment and study. YK and KO conducted the experiments. C-CS performed U-Th dating. All authors contributed to the discussion. RU analysed the results, generated figures and wrote the paper.

**Competing interests.**

The authors declare that they have no conflict of interest.

## Acknowledgements

We thank Eri Iwase, Satoru Mishima, Kosuke Masaka, Yuina Uechi, Tatsuki Matsuura, Masaaki Chinen (University of the Ryukyus) and Ryuji Asami (Tohoku University) for supporting the preparation and isotopic measurements, and Tsai-Luen Yu for U-Th dating. We also thank Kazuaki Higashi for the assistance with field work.

## Financial support

This work was supported by JSPS KAKENHI Grant Numbers 15H01729; 16H02235; 17KK0012; 18H03794. This study was also supported by the University of the Ryukyus Research Promotion Grant. U-Th dating was provided by grants from the Science Vanguard Research Program of the Ministry of Science and Technology (108-2119-M-002-012 to C.-C.S.) and the Higher Education Sprout Project of the Ministry of Education, Taiwan, ROC (108L901001 to C.-C.S.)

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

  **Tables**

**Table 1: Isotopic compositions of inclusion waters and calcite in HNS1 stalagmite for the 105°C experiment**

| ID | depth (mm) | Weight (mg) | $\delta^{18}O_{ca}$ (‰, VPDB) | water content (wt.%) | $\delta^{18}O_{fi}$ (‰, VSMOW) | $\delta D_{fi}$ (‰, VSMOW) | Drying time (h) | $\Delta\delta^{18}O_{fi}$ (‰, VSMOW) | $\Delta\delta D_{fi}$ (‰, VSMOW) |
|---|---|---|---|---|---|---|---|---|---|
| A | 74.2-76.8 | 151 | -5.29 | 0.14 | -5.88 | -34.25 | 0.1 | 0.00 | 0.0 |
| | | 97 | -5.29 | 0.15 | -5.85 | -37.02 | 1.0 | 0.03 | -2.8 |
| | | 100 | -5.29 | 0.14 | -5.51 | -35.72 | 7.0 | 0.37 | -1.5 |
| | | 114 | -5.29 | 0.26 | -5.00 | -31.45 | 18.0 | 0.88 | 2.8 |
| | | 51 | -5.29 | 0.10 | -5.91 | -36.32 | 33.5 | -0.03 | -2.1 |
| | | 97 | -5.29 | 0.14 | -4.79 | -32.70 | 60.0 | 1.09 | 1.5 |
| | | | | | | | | | |
| B | 83.3-86.3 | 54 | -5.53 | 0.12 | -5.96 | -35.88 | 0.1 | 0.00 | 0.0 |
| | | 91 | -5.53 | 0.16 | -4.84 | -33.69 | 15.0 | 1.12 | 2.2 |
| | | 75 | -5.53 | 0.08 | -4.88 | -34.99 | 68.0 | 1.08 | 0.9 |
| | | | | | | | | | |
| C | 102.0-105.9 | 128 | -5.18 | 0.06 | -6.41 | -33.59 | 0.1 | 0.00 | 0.0 |
| | | 193 | -5.18 | 0.09 | -6.19 | -36.90 | 15.5 | 0.22 | -3.3 |
| | | 183 | -5.18 | 0.09 | -5.30 | -32.90 | 16.0 | 1.10 | 0.7 |
| | | 158 | -5.18 | 0.09 | -5.87 | -33.73 | 44.0 | 0.54 | -0.1 |
| | | | | | | | | | |
| D* | 106.3-109.0 | 151 | -5.73 | 0.07 | -6.04 | -38.94 | 1.0 | 0.00 | 0.0 |
| | | 156 | -5.73 | 0.16 | -5.93 | -36.87 | 4.0 | 0.11 | 2.1 |
| | | 160 | -5.73 | 0.06 | -5.29 | -34.63 | 20.0 | 0.75 | 4.3 |
| | | 192 | -5.73 | 0.08 | -5.56 | -38.70 | 42.0 | 0.48 | 0.2 |
| | | | | | | | | | |
| E | 130.5-133.4 | 91 | -5.31 | 0.23 | -6.09 | -32.68 | 0.1 | 0.00 | 0.0 |
| | | 58 | -5.31 | 0.10 | -5.68 | -34.24 | 7.5 | 0.42 | -1.6 |
| | | 56 | -5.31 | 0.23 | -5.60 | -34.04 | 22.0 | 0.50 | -1.4 |
| | | 70 | -5.31 | 0.19 | -5.54 | -34.37 | 43.5 | 0.55 | -1.7 |

| | | | | | | | | | |
|---|---|---|---|---|---|---|---|---|---|
| | | 63 | -5.31 | 0.18 | -5.21 | -33.58 | 68.0 | 0.88 | -0.9 |

\* The initial value for sample D was one hour of drying.

**Table 2: Isotopic compositions of inclusion waters and calcite in HNS1 stalagmite for the 25°C experiment**

| ID | depth (mm) | Weight (mg) | $\delta^{18}O_{ca}$ (‰, VPDB) | water content (wt.%) | $\delta^{18}O_{fi}$ (‰, VSMOW) | $\delta D_{fi}$ (‰, VSMOW) | Drying time (h) | $\Delta\delta^{18}O_{fi}$ (‰, VSMOW) | $\Delta\delta D_{fi}$ (‰, VSMOW) |
|---|---|---|---|---|---|---|---|---|---|
| F | 95.4-98.6 | 176 | -5.52 | 0.10 | -5.87 | -33.01 | 0.0 | 0.0 | 0.0 |
| | | 155 | -5.52 | 0.07 | -6.39 | -34.91 | 19.0 | -0.5 | -1.9 |
| | | 169 | -5.52 | 0.06 | -6.13 | -32.99 | 42.5 | -0.3 | 0.0 |
| | | 111 | -5.52 | 0.15 | -6.40 | -34.58 | 71.5 | -0.5 | -1.6 |
| | | | | | | | | | |
| G | 117.0-119.4 | 76 | -5.35 | 0.22 | -5.95 | -32.67 | 0.0 | 0.00 | 0.0 |
| | | 90 | -5.35 | 0.13 | -5.96 | -34.24 | 6.0 | -0.01 | -1.6 |
| | | 86 | -5.35 | 0.22 | -5.85 | -33.49 | 22.5 | 0.10 | -0.8 |
| | | 79 | -5.35 | 0.16 | -6.65 | -35.98 | 55.0 | -0.70 | -3.3 |
| | | 92 | -5.35 | 0.16 | -6.01 | -34.85 | 73.5 | -0.06 | -2.2 |
| | | | | | | | | | |
| H | 120.0_123.5 | 151 | -5.38 | 0.08 | -5.93 | -35.17 | 0.0 | 0.00 | 0.0 |
| | | 116 | -5.38 | 0.06 | -6.15 | -34.31 | 27.0 | -0.23 | 0.9 |
| | | 155 | -5.38 | 0.19 | -6.44 | -33.93 | 50.5 | -0.51 | 1.2 |
| | | 119 | -5.38 | 0.11 | -5.79 | -33.39 | 72.0 | 0.13 | 1.8 |
| | | | | | | | | | |
| I | 148.6-152.2 | 84 | -4.94 | 0.08 | -6.41 | -34.98 | 0.0 | -0.12 | -0.6 |
| | | 113 | -4.94 | 0.14 | -6.18 | -33.81 | 0.0 | 0.12 | 0.6 |
| | | 107 | -4.94 | 0.25 | -5.94 | -33.57 | 6.0 | 0.35 | 0.8 |
| | | 95 | -4.94 | 0.31 | -5.79 | -30.86 | 23.0 | 0.51 | 3.5 |
| | | 71 | -4.94 | 0.15 | -6.77 | -36.49 | 43.5 | -0.48 | -2.1 |
| | | | | | | | | | |
| J | 177.9-180.6 | 92 | -5.30 | 0.16 | -6.44 | -34.59 | 0.0 | 0.00 | 0.0 |
| | | 134 | -5.30 | 0.18 | -5.73 | -32.05 | 52.5 | 0.71 | 2.5 |
| | | 84.7 | -5.30 | 0.16 | -5.83 | -32.23 | 77.0 | 0.61 | 2.4 |

**Table 3: Thorium-230 dating results of stalagmite HSN1 from Hoshino Cave**

| Depth (mm) | Weight (g) | $^{238}U$ (ppb) | $^{232}Th$ (ppt) | $\delta^{234}U$ (measured[a]) | $[^{230}Th/^{238}U]$ (activity[c]) | $[^{230}Th/^{232}Th]$ (ppm[d]) | Age (uncorrected) | Age (yr BP[c,e]) | $\delta^{234}U_{initial}$ (corrected [b]) |
|---|---|---|---|---|---|---|---|---|---|
| 75.0 | 0.2972 | 164.20± 0.46 | 2.2± 1.6 | 61.8± 3.3 | 0.06145±0.00047 | 76286± 54629 | 6497± 55 | 6429± 55 | 63.0± 3.0 |
| 190.0 | 0.2601 | 137.39± 0.26 | 3.7± 1.8 | 58.1± 2.1 | 0.06729±0.00041 | 41683± 20343 | 7160± 48 | 7092± 48 | 59.3±2.1 |

Chemical analyses were performed in March 2017. Analytical errors are 2σ of the mean.

[a] $[^{238}U] = [^{235}U] \times 137.818$ (±0.65 ‰) (Hiess et al., 2012); $\delta^{234}U = ([^{234}U/^{238}U]_{activity} - 1) \times 1000$.

[b] $\delta^{234}U_{initial}$ corrected was calculated based on $^{230}Th$ age (T), i.e., $\delta^{234}U_{initial} = \delta^{234}U_{measured} \times e^{\lambda_{234} \times T}$, and T is the corrected age.

[c] $[^{230}Th/^{238}U]$ activity $= 1 - e^{-\lambda_{230} \times T} + (\delta^{234}U_{measured} / 1000) [\lambda_{230} / (\lambda_{230} - \lambda_{234})] (1 - e^{-(\lambda_{230} - \lambda_{234}) T})$, where T is the age. The decay constants are $9.1705 \times 10^{-6}$ yr$^{-1}$ for $^{230}Th$, $2.8221 \times 10^{-6}$ yr$^{-1}$ for $^{234}U$ and $1.55125 \times 10^{-10}$ yr$^{-1}$ for $^{238}U$.

[d] The degree of detrital $^{230}Th$ contamination is indicated by the $[^{230}Th / ^{232}Th]$ atomic ratio instead of the activity ratio.

[e] Age corrections for samples were calculated using an estimated atomic $^{230}Th/^{232}Th$ ratio of $4 \pm 2$ ppm. These are the values for a material at secular equilibrium, with a crustal $^{232}Th/^{238}U$ of 3.8 with an arbitrary uncertainty of 50%. BP indicates Before Present, where Present is defined as 1950 CE.

**Figures**

**Figure 1: Photographs of HSN1 stalagmite.**

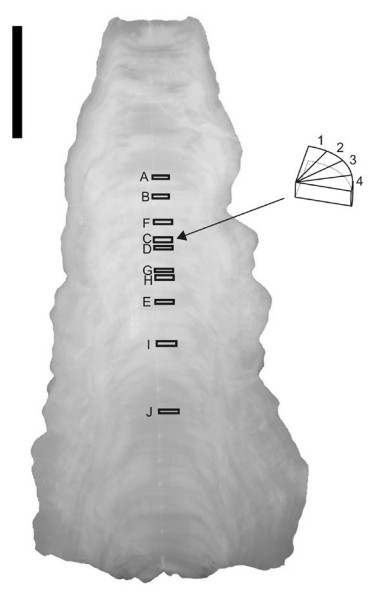

Half section of the HSN1 stalagmite from Hoshino Cave. The horizontal bar represents 5 cm. The positions of samples (A-J) used for the experiments are shown as bold black box. An example of wedge-shaped sub-samples from a layer is illustrated schematically.

**Figure 2: Changes in isotopic compositions of inclusion water with increasing drying time.**

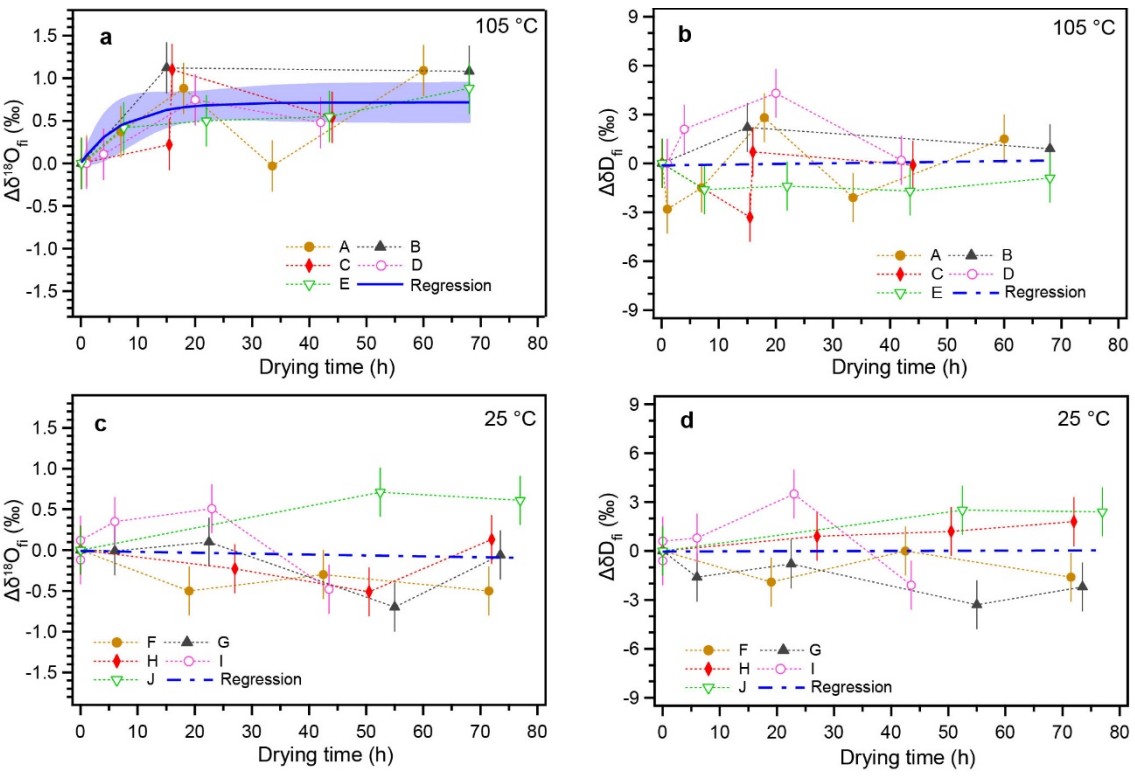

**a)** Changes in the $\delta^{18}O$ value of inclusion water from the initial value ($\Delta\delta^{18}_{fi(t)}$) with time under 105°C heating. Time on the x-axis indicates drying time in addition to the standard 17-hour drying process at room temperature. **b)** The same as **a)** but for $\delta D_{fi}$. **c)** Changes in the $\Delta\delta^{18}_{fi(t)}$ value over time at room temperature (25°C). **d)** The same as **c)** but for $\delta D_{fi}$. The error bar of the data point represents the analytical reproducibility (1σ). The regression curve (blue) in panel **a)** was calculated with IGOR (Wave Metrics Inc.) software to fit a function ($\Delta\delta^{18}_{fi(t)} = A\,(1 - \exp(-k\ t))$) (A = 0.71: k = 0.14). The blue shade indicates the 95% confidence interval. Blue dashed lines in panels **b-d** indicate linear regression lines (the correlations are insignificant).

**Figure 3: Changes in water content with increasing drying time.**

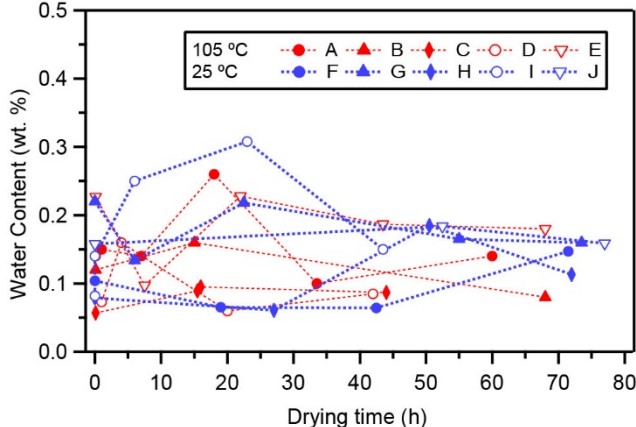

Changes in water content (in wt.%) of each subsample with time (in hour). Symbols indicate the 105°C experiment (red, layer A-E) and 25°C drying experiment (blue, layer F-J).

**Figure 4: D-O plot of inclusion water.**

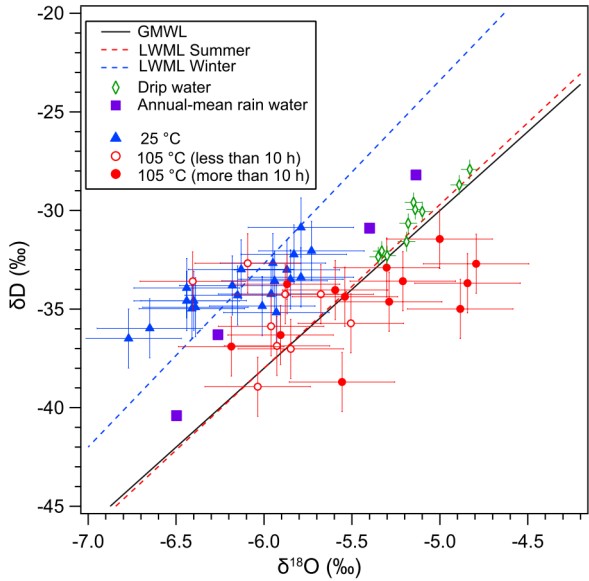

The blue triangles indicate the isotopic compositions of the fluid inclusion in the 25°C drying experiment. The red circles indicate the isotopic compositions of the fluid inclusion in the 105°C drying experiment for < 10 hours (open red circles) and >10 hours (solid red circles). The black line indicates GMWL. Local meteoric water lines based on present-day raindata from Okinawa-jima Island for the summer season (red dotted line) and the winter season (blue dotted line) are also shown.

Purple solid square indicates precipitation-weighted annual mean values (2009-2012) of the rain water on Okinawa-jima Island. Green open diamond indicates present-day observation of dripwater in Hoshino cave.

**Figure 5: Schematic illustration of isotopic re-equilibration between inclusion water and stalagmite.**

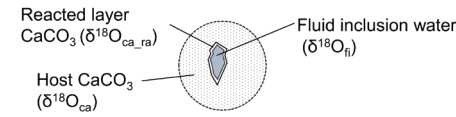

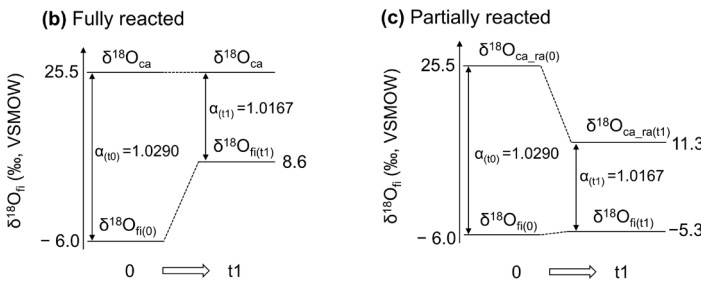

(**a**) Schematic illustration of a water-filled fluid inclusion and surrounding $CaCO_3$. (**b**) Changes in the isotopic composition for the fully reacted hypothesis, in which the $\delta^{18}O_{fi}$ value changes in response to equilibration with an infinite amount of host

$CaCO_3$. In this case, the sample at room temperature (25°C at time of 0) is heated to 105°C and reaches a new isotopic equilibrium at time $t_1$. (**c**) The same as (**b**) but for the partially reacted hypothesis, in which the $\delta^{18}O_{fi}$ value changes in response to equilibration with a limited amount of $CaCO_3$. Note that the calcite-water fractionation factor is the same for both hypotheses.