# Peer review of "Experimental evaluation of oxygen isotopic exchange between inclusion water and host calcite in speleothems"

_Climate of the Past, 2019_

## Referee Comment (RC1) · Anonymous Referee #1 · 30 Aug 2019

General comments:

The paper experimentally investigates the post-depositional oxygen isotope exchange of fluid inclusion water. As an increasing number of laboratories are developing and applying techniques for fluid inclusion analysis, a sound understanding of the related isotope signals and potential limitations is urgently needed. The constraints provided by this study are therefore very valuable and show that in most cases also the fluid inclusion d18O signal may reflect the drip water at time of enclosure.

The paper is well structured and written. The used technique is clearly described, all necessary data for discussion are given, and the interpretation is based on the authors'

genuine data. The topic is well within the scope of Climate of the Past as it addresses an emerging proxy with high paleoclimatic significance.

Specific comments:

Introduction: -line 36-37: Cave dripwaters... usually close to the d18O of local rain. Is it? Or is it close to the infiltration-weighted mean of the rain? A literature reference may be sufficient (e.g., Baker et al., 2019, Nat. Commun.)

-line 42: ...suggests a relatively stable value for the temperature dependence of d18Oca... I would be a bit more cautious here. Mühlinghaus et al. (2009) found that the change of the calcite d18O with temperature has a certain relation to the drip interval which is expressed differently at different cave temperatures. It is relatively limited at 10°C (-0.22 to -0.26 ‰·°C), but quite substantial at 25°C (-0.21 to -0.35 ‰·°C) for the modelled drip intervals.

Methods -section 2.1: Did you take the speleothem samples for fluid inclusions at the growth axis or off-axis? The position relative to the axis may have an influence on the water content and may also be interesting for the discussion section and Fig.3.

Results and discussion: -lines 137-138: you state that the observed increase is due to exchange between inclusion water and calcite. Could calcite dissolution or new calcite precipitation related to a change in the saturation state following the increased temperature also play a role?

-lines 144-145: Leakage may also be influenced by the fabric of the stalagmite and be different in inclusions with e.g., large columnar crystals compared to dendritic parts. What is the fabric of the investigated pieces of HSN1? Also the following statements in lines 148-152 should be rephrased considering potentially different behaviour of stalagmites with different fabric and micro-structure. Lacking larger data sets of various stalagmites I would be hesitant to generalize. Still the statements are ok for the analysed sample but should be written in a way that a generalisation for the leakage aspect

is avoided.

-lines 168-176: is the deuterium excess indeed a (better) indicator for oxygen isotope exchange compared to the closeness to the LMWL?

-lines 213-215: This sentence may be misunderstood and should be slightly rephrased. Whereas the rate constant of the isotope exchange reaction only varies with temperature, the number of transferred isotopes varies with the temporal evolution of the isotope ratios of the end members.

Fig.1: Please indicate in the figure where the samples were taken for the analysis.

Fig.4: Do you have d18O and dD values of modern drip water from the cave? Or could you alternatively calculate the infiltration-weighted mean of the rainfall? It may illustrate additionally the shift between the 105°C samples and the room-temperature reference that should be close to the dripwater.

Typos: -line 60: "These data suggest *an* isotopic exchange of . . .." or just without the current "the" -line 131: ". . .of inclusion water *are* shown as deviation from . . ." -lines 141-142: either "there is little hydrogen in the calcite" or "there is no significant hydrogen reservoir in the calcite"

---

## Referee Comment (RC2) · Anonymous Referee #2 · 6 Sep 2019

This contribution by Uemura et al. presents a series of experiment concerning the diagenetic alteration of fluid inclusion water in speleothem (stalagmite) calcite.

Over the past decades, fluid inclusion oxygen and hydrogen isotope analysis has become accepted to be a very useful addition to the toolbox of geochemical proxies applied on speleothem (stalagmite) climate archives. As stalagmite fluid inclusion isotope data are believed to represent the isotope composition of rainwater back in time, they potentially provide the warm-climate equivalent of the famous Ice Core isotope records. There are complications, however. One major issue is the potential post-depositional oxygen isotope exchange between the fluid inclusion water and host calcite, that cause

the fluid inclusion isotope measurement to represent a diagenetic, rather than a rainwater d18O value. Uemura et al. study this process by heating stalagmite material to 105 degrees C, in an attempt to force diagenetic (thermal) alteration of fluid inclusion water. Their results show that such alteration takes place, but has a rather limited effect on the d18O value of the fluid inclusions only. It is demonstrated how this likely reflects isotope exchange with a very thin layer of CaCO3 around each fluid inclusion. Extrapolation of these results to reasonable natural (glacial-interglacial) variation in temperature that stalagmites are exposed to, suggest that the effects of such limited temperature change on diagenetic alteration of the fluid inclusion water is negligible.

In summary, I believe this is an interesting experiment that gives, for the first time, quantitative insight in the extent of diagenetic alteration of fluid inclusions. The results line up well with several published fluid inclusion stable isotope records that indicate that such diagenetic alteration is not a common process in stalagmites. The ms is well-written, and the experimental results logically explained.

I could phrase some possible criticism on the experiment concerning the following aspects of the studied material and experimental set up:

1) I find no age model of the studied stalagmite, so that it is unclear this this is submodern or ancient material (or anything in between). Not having the ages leaves some uncertainty to what extent the fluid inclusion isotope data should coincide with the meteoric waterline presented in the ms.

2) If I read the ms correctly the high temperature experiment is not performed on samples of the same age (layer) as the low temperature experiment. It would perhaps have been more elegant if that would have been the case. This is by no means a fatal problem since the significant isotope shift through heating takes place in samples that are from a single layer, so the central outcome of this study is fully supported by the data anyhow.

3) What strikes me in the low temperature d2H d18O cross plot is that while the authors state the data to lie between the winter and summer meteoric water lines, the supposedly original fluid inclusion isotope values seem somewhat biased towards the winter rainfall composition to me. If I'm correct, this could have several reasons: 1 Winter rainfall dominating cave recharge at this site. 2 a small isotope artefact in the fluid inclusion isotope data 3 the calculated seasonal meteoric water lines are based on data from another island, and thus may not be entirely representative for the study site. 4 The stalagmite could be somewhat older, and precipitated under a different from modern rainfall regime.

While I'd be interested to hear if the authors agree with my observation, I realize that this latter point can probably not be clarified because not all the information on study site and cave environment (particularly drip water isotope ratios?) are available. I think it is fair to say that such detailed considerations are beyond the scope of the present study. Bottom line is that in this study Uemura et al show clearly to which extent heating experiments are able to affect the isotope ratios of fluid inclusions in stalagmite calcite. I believe this improves our insight in the potential of fluid inclusion isotope analysis in stalagmite records, and thus provides a valuable contribution to the rapidly developing field of speleothem climate science.

Some more detailed comments: 1) Your statement in line 77-79 may require a reference. 2) Where you describe the reproducibility of the fluid inclusion isotope analysis it is not clear what this is based on. Repeated analyses of water samples? replicate analyses of fluid inclusion water? please specify. 3) similar question for the Gasbench isotope analyses. Are the reported uncertainties the sample-internal standard deviations, or based on longer series of carbonate standards? 4)in line 175 you claim that isotopic exchange results in lower d-excess values. Technically speaking this is only the case if the exchange takes place at higher temperatures than the original precipitation temperature. This could essentially go the other way around as well, so it is not exclusively the lower d-excess values that could be indicative for oxygen isotope exchange. Rather, any deviation from the MWL could potentially signal oxygen isotope

exchange under changing temperatures.

I found no further flaws or technical details that must be changed. I therefore would suggest that this can be published with just a few minor (technical) corrections.

---

## Author Comment (AC1) · 18 Oct 2019

Response to anonymous Referee #1 We thank the anonymous Referee #1for the detailed and constructive comments. Below, we respond to the comments in order (Referee's comments are marked with "[#R1-x]". Our responses are marked with "*").

[#R1-1] "General comments: The paper experimentally investigates the post-depositional oxygen isotope exchange of fluid inclusion water. As an increasing number of laboratories are developing and applying techniques for fluid inclusion analysis, a sound understanding of the related isotope signals and potential limitations is urgently needed. The constraints provided by this study are therefore very valuable and show

that in most cases also the fluid inclusion $\delta$18O signal may reflect the drip water at time of enclosure. The paper is well structured and written. The used technique is clearly described, all necessary data for discussion are given, and the interpretation is based on the authors'genuine data. The topic is well within the scope of Climate of the Past as it addresses an emerging proxy with high paleoclimatic significance."

*We appreciate hearing your positive view of our manuscript.

[#R1-2] "Specific comments: Introduction: -line 36-37: Cave dripwaters. . . usually close to the d18O of local rain. Is it? Or is it close to the infiltration-weighted mean of the rain? A literature reference may be sufficient (e.g., Baker et al., 2019, Nat. Commun.)"

*Corrected. We also added the reference you suggested.

[#R1-3] '-line 42: . . .suggests a relatively stable value for the temperature dependence of d18Oca. . . I would be a bit more cautious here. Mühlinghaus et al. (2009) found that the change of the calcite $\delta$18O with temperature has a certain relation to the drip interval which is expressed differently at different cave temperatures. It is relatively limited at 10degC(-0.22 to -0.26 ‰), but quite substantial at 25degC (-0.21 to -0.35 ‰) for the modelled drip intervals."

* We think that our expression "relatively stable" is too ambiguous. We intended to say the temperature reconstruction is possible. In fact, Mühlinghaus et al. (2009) wrote: "With increasing drip interval the temperature dependence changes due to the buffer-ing and mixing processes but is always within the range of -0.20 and -0.34‰°C. This implies that the interpretation of d18O variations in terms of temperature is still possible for stalagmites grown under conditions of isotopic disequilibrium, if the stalagmite was fed by a relatively constant drip rate." Of course, the slope is not perfectly constant but still possible with uncertainties. In order to clarify the meaning, we removed the expression "relatively stable". Instead, we added exact values of the slope (-0.20 and -0.34permil/$^\circ$C) in the sentence.

[#R1-4] "Methods -section 2.1: Did you take the speleothem samples for fluid inclusions at the growth axis or off-axis? The position relative to the axis may have an influence on the water content and may also be interesting for the discussion section and Fig.3."

*We took the samples from the growth axis. Then, a fan shaped sub-sample was symmetrically divided into 3-6 pieces. Thus, the position does not matter. A schematic illustration will be added to explain the shape of sample in Figure 1 (please also refer to [#R1-9]).

[Please refer to the Supplemental PDF for the cut out of revised version of Fig.1 ]

[#R1-5] "Results and discussion: -lines 137-138: you state that the observed increase is due to exchange between inclusion water and calcite. Could calcite dissolution or new calcite precipitation related to a change in the saturation state following the increased temperature also play a role?"

* We think that dissolution or new calcite precipitation in fluid inclusions could play a role. It is, however, difficult to estimate such effects because it depends on pH and the amount of $CO_2$ in inclusions. In the case of our experiment, the new calcite precipitation did not occur in the inclusions because the $\delta 18O$ value of water is expected to be lower if the new calcite, whose $\delta 18O$ value is higher than that of water, formed inside the inclusions. This is opposite to the result of heating experiment. In the case of internal calcite dissolution, the $\delta 18O$ value of water, will be changed through the isotopic exchange reaction between the bicarbonate in the solution and the water reservoir. Thus, essentially, it would not different from the case of re-equilibration between calcite and water. We will add sentences about these possibilities in the revised MS.

[#R1-6] "-lines 144-145: Leakage may also be influenced by the fabric of the stalagmite and be different in inclusions with e.g., large columnar crystals compared to dendritic parts. What is the fabric of the investigated pieces of HSN1? Also the following statements in lines 148-152 should be rephrased considering potentially different behavior of stalagmites with different fabric and micro-structure. Lacking larger data sets of various stalagmites, I would be hesitant to generalize. Still the statements are ok for the analyzed sample but should be written in a way that a generalization for the leakage aspect is avoided. "

*Yes, the characteristics of leakage would be different for the different stalagmites. We revised the sentences to describe the potentially different behavior of stalagmites with different fabric and micro-structure in Section 3.2. A description about the fabric of HSN1, open columnar structure, was also added in Section 2.1 "Speleothem samples".

[#R1-7] "-lines 168-176: is the deuterium excess indeed a (better) indicator for oxygen isotope exchange compared to the closeness to the LMWL?"

*The d-excess is equivalent to the "closeness to the LMWL" when the slope of LMWL is close to 8. We think that the d-excess has advantage of being quantitative (i.e., easy to show the difference with numbers). In addition, we think that this sentence is misleading because logically the opposite direction (higher d-excess) may occur (please also refer to [#R2-8]). Thus, we deleted this sentence and will revise it as follows. "We should note that the d-excess value could become higher if the exchange takes place at lower temperatures than the original precipitation temperature. Therefore, the oxygen isotope exchange under changing temperatures may cause any slight deviation from the LMWL."

[#R1-8] "-lines 213-215: This sentence may be misunderstood and should be slightly rephrased. Whereas the rate constant of the isotope exchange reaction only varies with temperature, the number of transferred isotopes varies with the temporal evolution of the isotope ratios of the end members."

* The sentence was rephrased.

[#R1-9] "Fig.1: Please indicate in the figure where the samples were taken for the analysis."

* We will add the positions of the samples and shape of subsample in Figure 1 (please

also refer to [#R1-4]).

[#R1-10] "Fig.4: Do you have $\delta$18O and $\delta$D values of modern drip water from the cave? Or could you alternatively calculate the infiltration-weighted mean of the rainfall? It may illustrate additionally the shift between the 105°C samples and the room-temperature reference that should be close to the dripwater."

*The modern dripwater data will be added to Fig. 4. The HSN1 stalagmite is mid-Holocene sample (6429±55 and 7092±48 years BP) (please also refer to [#R2-2] ). Thus, the rain water isotope ratio is likely different from modern rainfall (please also refer to [#R2-4]).

[#R1-11] "Typos: -line 60: "These data suggest *an* isotopic exchange of . . .." or just without the current "the" -line 131: ". . .of inclusion water *are* shown as deviation from . . ."-lines 141-142: either "there is little hydrogen in the calcite" or "there is no significant hydrogen reservoir in the calcite"

* Corrected. We thank you again for your comments and suggestions.

Please also note the supplement to this comment:
https://www.clim-past-discuss.net/cp-2019-79/cp-2019-79-AC1-supplement.pdf

―――――――――――――――

---

## Author Response (AR1)

**Response to anonymous Referee #1**

We thank the anonymous Referee #1 for the detailed and constructive comments. Below, we respond to the comments in order (Referee's comments are marked with "[**#R1-x**]". Our responses are marked with "\*").

**[#R1-1]** "General comments: The paper experimentally investigates the post-depositional oxygen isotope exchange of fluid inclusion water. As an increasing number of laboratories are developing and applying techniques for fluid inclusion analysis, a sound understanding of the related isotope signals and potential limitations is urgently needed. The constraints provided by this study are therefore very valuable and show that in most cases also the fluid inclusion  $\delta^{18}$ O signal may reflect the drip water at time of enclosure. The paper is well structured and written. The used technique is clearly described, all necessary data for discussion are given, and the interpretation is based on the authors'genuine data. The topic is well within the scope of Climate of the Past as it addresses an emerging proxy with high paleoclimatic significance."

**\*We appreciate hearing your positive view of our manuscript.**

**[#R1-2]** "Specific comments: Introduction: -line 36-37: Cave dripwaters. . . usually close to the d18O of local rain. Is it? Or is it close to the infiltration-weighted mean of the rain? A literature reference may be sufficient (e.g., Baker et al., 2019, Nat. Commun.)"

**\*Corrected. We also added the reference you suggested.**

**[#R1-3]** '-line 42: . . .suggests a relatively stable value for the temperature dependence of d18Oca. . . I would be a bit more cautious here. Mühlinghaus et al. (2009) found that the change of the calcite  $\delta^{18}$ O with temperature has a certain relation to the drip interval which is expressed differently at different cave temperatures. It is relatively limited at 10°C (-0.22 to -0.26 ‰C), but quite substantial at 25C (-0.21 to -0.35 ‰C) for the modelled drip intervals."

\* We think that our expression "relatively stable" is too ambiguous. We intended to say the temperature reconstruction is possible. In fact, Mühlinghaus et al. (2009) wrote: "*With increasing drip interval the temperature dependence changes due to the*  buffering and mixing processes but is always within the range of -0.20 and -0.34‰/°C. This implies that the interpretation of  $d^{18}O$  variations in terms of temperature is still possible for stalagmites grown under conditions of isotopic disequilibrium, if the stalagmite was fed by a relatively constant drip rate." Of course, the slope is not perfectly constant but still possible with uncertainties. In order to clarify the meaning, we removed the expression "relatively stable". Instead, we added exact values of the slope (-0.20 and -0.34permil/°C) in the sentence.

**[#R1-4]** "Methods -section 2.1: Did you take the speleothem samples for fluid inclusions at the growth axis or off-axis? The position relative to the axis may have an influence on the water content and may also be interesting for the discussion section and Fig.3."

\*We took the samples from the growth axis. Then, a fan shaped sub-sample was symmetrically divided into 3-6 pieces. Thus, the position does not matter. A schematic illustration was added to explain the shape of sample in Figure 1 (please also refer to [#R1-9]).

[Revised version of Fig.1]

**[#R1-5]** "Results and discussion: -lines 137-138: you state that the observed increase is due to exchange between inclusion water and calcite. Could calcite dissolution or new calcite precipitation related to a change in the saturation state following the increased

temperature also play a role?"

\* We think that dissolution or new calcite precipitation in fluid inclusions could play a role. It is, however, difficult to estimate such effects because it depends on pH and the amount of CO2 in inclusions. In the case of our experiment, the new calcite precipitation did not occur in the inclusions because the  $\delta^{18}$ O value of water is expected to be lower if the new calcite, whose  $\delta^{18}$ O value is higher than that of water, formed inside the inclusions. This is opposite to the result of heating experiment. In the case of internal calcite dissolution, the  $\delta^{18}$ O value of water, will be changed through the isotopic exchange reaction between the bicarbonate in the solution and the water reservoir. Thus, essentially, it would not different from the case of reequilibration between limited amount of calcite and water. We added sentences about these possibilities in Section 3.1.

**[#R1-6]** "-lines 144-145: Leakage may also be influenced by the fabric of the stalagmite and be different in inclusions with e.g., large columnar crystals compared to dendritic parts. What is the fabric of the investigated pieces of HSN1? Also the following statements in lines 148-152 should be rephrased considering potentially different behavior of stalagmites with different fabric and micro-structure. Lacking larger data sets of various stalagmites, I would be hesitant to generalize. Still the statements are ok for the analyzed sample but should be written in a way that a generalization for the leakage aspect is avoided. "

\*Yes, the characteristics of leakage would be different for the different stalagmites. We revised the sentences to describe the potentially different behavior of stalagmites with different fabric and micro-structure in Section 3.2. A description about the fabric of HSN1, open columnar structure, was also added in Section 2.1 "Speleothem samples".

**[#R1-7]** "-lines 168-176: is the deuterium excess indeed a (better) indicator for oxygen isotope exchange compared to the closeness to the LMWL?"

\*The d-excess is equivalent to the "closeness to the LMWL" when the slope of LMWL is close to 8. We think that the d-excess has advantage of being quantitative (i.e., easy to show the difference with numbers). In addition, we realize that this

sentence was misleading because logically the opposite direction (higher d-excess) may occur (please also refer to [#R2-8]). Thus, we deleted this sentence and revised it as follows.

"We should note that the d-excess value could become higher if the exchange takes place at lower temperatures than the original precipitation temperature. Therefore, the oxygen isotope exchange under changing temperatures may cause any slight deviation from the LMWL."

**[#R1-8]** "-lines 213-215: This sentence may be misunderstood and should be slightly rephrased. Whereas the rate constant of the isotope exchange reaction only varies with temperature, the number of transferred isotopes varies with the temporal evolution of the isotope ratios of the end members."

\* The sentence was rephrased.

[#R1-9] "Fig.1: Please indicate in the figure where the samples were taken for the analysis."

\* We added the positions of the samples and an example of the shape of subsample in Figure 1 (please also refer to [#R1-4]).

*[#R1-10]* "Fig.4: Do you have  $\delta^{18}$ O and  $\delta$ D values of modern drip water from the cave? Or could you alternatively calculate the infiltration-weighted mean of the rainfall? It may illustrate additionally the shift between the 105°C samples and the room-temperature reference that should be close to the dripwater."

\*The modern dripwater data were added to Fig. 4. The HSN1 stalagmite is mid-Holocene sample (6429±55 and 7092±48 years BP) (please also refer to [#R2-2]). Thus, the rainwater isotope ratio is different from modern rainfall (please also refer to [#R2-4]). We revised Fig. 4 and its caption. Related short discussions were also added.

[#R1-11] "Typos: -line 60: "These data suggest \*an\* isotopic exchange of . . .. " or just

without the current "the" -line 131: ". . . of inclusion water \*are\* shown as deviation from . . . "-lines 141-142: either "there is little hydrogen in the calcite" or "there is no significant hydrogen reservoir in the calcite"

**\* Corrected.**

We thank you again for your comments and suggestions.

**Response to anonymous Referee #2**

We thank the anonymous Referee #2 for the detailed and constructive comments. Below, we respond to the comments in order. Referee's comments are marked with "#" and add sequential numbers ([R2-x]). Our responses are marked with "\*".

[**#R2-1**] "This contribution by Uemura et al. presents a series of experiment concerning the diagenetic alteration of fluid inclusion water in speleothem (stalagmite) calcite. Over the past decades, fluid inclusion oxygen and hydrogen isotope analysis has become accepted to be a very useful addition to the toolbox of geochemical proxies applied on speleothem (stalagmite) climate archives. As stalagmite fluid inclusion isotope data are believed to represent the isotope composition of rainwater back in time, they potentially provide the warm-climate equivalent of the famous Ice Core isotope records. There are complications, however. One major issue is the potential post-depositional oxygen isotope exchange between the fluid inclusion water and host calcite, that cause the fluid inclusion isotope measurement to represent a diagenetic, rather than a rain water  $\delta^{18}$ O value. Uemura et al. study this process by heating stalagmite material to 105 degrees C, in an attempt to force diagenetic (thermal) alteration of fluid inclusion water. Their results show that such alteration takes place, but has a rather limited effect on the  $\delta^{18}O$  value of the fluid inclusions only. It is demonstrated how this likely reflects isotope exchange with a very thin layer of CaCO3 around each fluid inclusion. Extrapolation of these results to reasonable natural (glacial-interglacial) variation in temperature that stalagmites are exposed to, suggest that the effects of such limited temperature change on diagenetic alteration of the fluid inclusion water is negligible. In summary, I believe this is an interesting experiment that gives, for the first time, quantitative insight in the extent of diagenetic alteration of fluid inclusions. The results line up well with several published fluid inclusion stable isotope records that indicate that such diagenetic alteration is not a common process in stalagmites. The ms is well-written, and the experimental results logically explained."

**\*We appreciate hearing your positive view of our manuscript.**

**[#R2-2]** "I could phrase some possible criticism on the experiment concerning the following aspects of the studied material and experimental set up:

1) I find no age model of the studied stalagmite, so that it is unclear this this is sub modern or ancient material (or anything in between). Not having the ages leaves some uncertainty

to what extent the fluid inclusion isotope data should coincide with the meteoric waterline presented in the ms."

\*The HSN1 stalagmite is Holocene sample. We will add U-Th age data for the upper and lower layers sampled for this experiment as a new Table. The dates for the 75 mm and 190 mm layers are 6429±55 and 7092±48 years BP (before 1950), respectively. Accordingly, we added a short description about the dating in Methods and Results. We also added a coauthor (C-C. Shen) for the U-Th dating experiment.

Table 3: Thorium-230 dating results of stalagmite HSN1 from Hoshino Cave

| +. |                   |                   |                               |            |                                                       |                                                                        |                                                                    |                          |                                    |                                                                       |
|----|-------------------|-------------------|-------------------------------|------------|-------------------------------------------------------|------------------------------------------------------------------------|--------------------------------------------------------------------|--------------------------|------------------------------------|-----------------------------------------------------------------------|
|    | Depth 4
(mm) 0 | Weight +
(g) + | 238 U ↓
(ppb) ↓ | (ppt) +    | $\delta^{234}U +$
(measured a ) $\circ$ | [ 230 Th/ 238 U] +
(activity e ) - | [ 230 Th/ 232 Th] +
(ppm d ) + | Age +
(uncorrected) + | Age :
(yr BP c,e ) - | δ 234 U initial↓
(corrected b ) ₀ |
|    | 75.0 .            | 0.2972 .          | 164.20± 0.46 +                | 2.2± 1.6 » | 61.8± 3.3 .                                           | 0.06145±0.00047 .                                                      | $76286 \pm 54629  \diamond$                                        | 6497± 55 +               | 6429± 55 «                         | 63.0± 3.0 °                                                           |
|    | 190.0 -           | 0.2601 0          | 137.39± 0.26 -                | 3.7± 1.8 - | 58.1±2.1 +                                            | 0.06729±0.00041 .                                                      | 41683± 20343 »                                                     | 7160± 48 «               | 7092± 48 «                         | 59.3±2.1 +                                                            |

**[#R2-3]** "If I read the ms correctly the high temperature experiment is not performed on samples of the same age (layer) as the low temperature experiment. It would perhaps have been more elegant if that would have been the case. This is by no means a fatal problem since the significant isotope shift through heating takes place in samples that are from a single layer, so the central outcome of this study is fully supported by the data anyhow.

**\*Yes, the two experiments were not performed on the same sample. The sampling depths have been added in the revised version of Figure 1.**

**[#R2-4]** "3) What strikes me in the low temperature  $\delta^2 H \delta^{18}O$  cross plot is that while the authors state the data to lie between the winter and summer meteoric water lines, the supposedly original fluid inclusion isotope values seem somewhat biased towards the winter rainfall composition to me. If I'm correct, this could have several reasons: 1) Winter rainfall dominating cave recharge at this site. 2) a small isotope artefact in the fluid inclusion isotope data 3) the calculated seasonal meteoric water lines are based on data from another island, and thus may not be entirely representative for the study site. 4) The stalagmite could be somewhat older, and precipitated under a different from modern rainfall regime. While I'd be interested to hear if the authors agree with my observation, I realize that this latter point can probably not be clarified because not all the information

on study site and cave environment (particularly drip water isotope ratios?) are available. I think it is fair to say that such detailed considerations are beyond the scope of the present study. Bottom line is that in this study Uemura et al show clearly to which extent heating experiments are able to affect the isotope ratios of fluid inclusions in stalagmite calcite. I believe this improves our insight in the potential of fluid inclusion isotope analysis in stalagmite records, and thus provides a valuable contribution to the rapidly developing field of speleothem climate science.

\* We have added the age data for the HSN1 stalagmite (please also refer to [#R2-2]). The sample was grown during mid-Holocene and the rainwater isotope ratio is likely different from modern rainfall (corresponding to your hypothesis no. 4). We added modern drip water data in Hoshino cave in figure 4. Moreover, we also added the data of precipitation-weighted annual mean of rainwater on Okinawa-jima and added short discussions about the results in Section 3.3.

New Fig. 4

**[#R2-5]** Some more detailed comments: 1) Your statement in line 77-79 may require a reference.

**\*We added references.**

**[#R2-6]** 2) Where you describe the reproducibility of the fluid inclusion isotope analysis it is not clear what this is based on. Repeated analyses of water samples? Replicate analyses of fluid inclusion water? please specify.

\*The reproducibility is based on replicate analyses of fluid inclusion water. We revised the sentence to specify this.

**[#R2-7]** 3) similar question for the Gasbench isotope analyses. Are the reported uncertainties the sample-internal standard deviations, or based on longer series of carbonate standards?

\*The reproducibility is based on a long-term analysis (n=124 during the measurement period (Jan 2013-April 2014) of a carbonate standard, IAEA-CO-1. We measured a few "CO-1" for a batch of measurement to check the reproducibility.

**[#R2-8]** 4)in line 175 you claim that isotopic exchange results in lower d-excess values. Technically speaking this is only the case if the exchange takes place at higher temperatures than the original precipitation temperature. This could essentially go the other way around as well, so it is not exclusively the lower d-excess values that could be indicative for oxygen isotope exchange. Rather, any deviation from the MWL could potentially signal oxygen isotope exchange under changing temperatures.

\* Yes, this sentence is misleading. Logically the opposite direction (higher d-excess) may occur, as you suggested (please also refer to [#R1-7]). We deleted this sentence and revised it as follows.

"We should note that the d-excess value could become higher if the exchange takes place at lower temperatures than the original precipitation temperature. Therefore, the oxygen isotope exchange under changing temperatures may cause any slight deviation from the LMWL."

**[#R2-9]** I found no further flaws or technical details that must be changed. I therefore would suggest that this can be published with just a few minor (technical) corrections.

\*Thank you again for your constructive comments.

[revised manuscript text omitted]

| ID | depth
(mm)   | Weight
(mg) | δ 18 O ca
(‰,
VPDB) | water
content
(wt.%) | δ 18 O fi
(‰,
VSMOW) | δD fi
(‰,
VSMOW) | Drying
time (h) | Δδ 18 O fi
(‰,
VSMOW) | ΔδD fi
(‰,
VSMOW) |
|----|-----------------|----------------|-------------------------------------------------|----------------------------|--------------------------------------------------|-----------------------------------|--------------------|---------------------------------------------------|------------------------------------|
| А  | 74.2-76.8       | 151            | -5.29                                           | 0.14                       | -5.88                                            | -34.25                            | 0.1                | 0.00                                              | 0.0                                |
|    | -               | 97             | -5.29                                           | 0.15                       | -5.85                                            | -37.02                            | 1.0                | 0.03                                              | -2.8                               |
|    | -               | 100            | -5.29                                           | 0.14                       | -5.51                                            | -35.72                            | 7.0                | 0.37                                              | -1.5                               |
|    | -               | 114            | -5.29                                           | 0.26                       | -5.00                                            | -31.45                            | 18.0               | 0.88                                              | 2.8                                |
|    | -               | 51             | -5.29                                           | 0.10                       | -5.91                                            | -36.32                            | 33.5               | -0.03                                             | -2.1                               |
|    |                 | 97             | -5.29                                           | 0.14                       | -4.79                                            | -32.70                            | 60.0               | 1.09                                              | 1.5                                |
| В  | 83.3-86.3       | 54             | -5.53                                           | 0.12                       | -5.96                                            | -35.88                            | 0.1                | 0.00                                              | 0.0                                |
|    | -               | 91             | -5.53                                           | 0.16                       | -4.84                                            | -33.69                            | 15.0               | 1.12                                              | 2.2                                |
|    |                 | 75             | -5.53                                           | 0.08                       | -4.88                                            | -34.99                            | 68.0               | 1.08                                              | 0.9                                |
| С  | 102.0-          | 128            | -5.18                                           | 0.06                       | -6.41                                            | -33.59                            | 0.1                | 0.00                                              | 0.0                                |
|    | 105.9           | 193            | -5.18                                           | 0.09                       | -6.19                                            | -36.90                            | 15.5               | 0.22                                              | -3.3                               |
|    | -               | 183            | -5.18                                           | 0.09                       | -5.30                                            | -32.90                            | 16.0               | 1.10                                              | 0.7                                |
|    | -               | 158            | -5.18                                           | 0.09                       | -5.87                                            | -33.73                            | 44.0               | 0.54                                              | -0.1                               |
|    |                 |                |                                                 |                            |                                                  |                                   |                    |                                                   |                                    |
| D* | 106.3-
109.0 | 151            | -5.73                                           | 0.07                       | -6.04                                            | -38.94                            | 1.0                | 0.00                                              | 0.0                                |
|    |                 | 156            | -5.73                                           | 0.16                       | -5.93                                            | -36.87                            | 4.0                | 0.11                                              | 2.1                                |
|    |                 | 160            | -5.73                                           | 0.06                       | -5.29                                            | -34.63                            | 20.0               | 0.75                                              | 4.3                                |
|    |                 | 192            | -5.73                                           | 0.08                       | -5.56                                            | -38.70                            | 42.0               | 0.48                                              | 0.2                                |
|    |                 |                |                                                 |                            |                                                  |                                   |                    |                                                   |                                    |
| Е  | 130.5-
133.4 | 91             | -5.31                                           | 0.23                       | -6.09                                            | -32.68                            | 0.1                | 0.00                                              | 0.0                                |
|    |                 | 58             | -5.31                                           | 0.10                       | -5.68                                            | -34.24                            | 7.5                | 0.42                                              | -1.6                               |
|    |                 | 56             | -5.31                                           | 0.23                       | -5.60                                            | -34.04                            | 22.0               | 0.50                                              | -1.4                               |
|    |                 | 70             | -5.31                                           | 0.19                       | -5.54                                            | -34.37                            | 43.5               | 0.55                                              | -1.7                               |

| Table 1: Isotopic compositions of inclusion waters and calcite in HNS1 stalagmite for the 105°C experime | nt |
|----------------------------------------------------------------------------------------------------------|----|
|----------------------------------------------------------------------------------------------------------|----|

|  | 63 | -5.31 | 0.18 | -5.21 | -33.58 | 68.0 | 0.88 | -0.9 |
|--|----|-------|------|-------|--------|------|------|------|
|  |    |       |      |       |        |      |      |      |

\* The initial value for sample D was one hour of drying.

**435**

**Table 2: Isotopic compositions of inclusion waters and calcite in HNS1 stalagmite for the 25°C experiment**

| ID | depth       | Weight | δ 18 Oca | water   | $\delta^{18}O_{\rm fi}$ | δD fi | Drying | $\Delta \delta^{18} O_{\rm fi}$ | $\Delta \delta D_{\rm fi}$ |
|----|-------------|--------|---------------------|---------|-------------------------|------------------|--------|---------------------------------|----------------------------|
|    | (mm)        | (mg)   | (‰,                 | content | (‰,                     | (‰,              | time   | (‰,                             | (‰,                        |
|    |             |        | VPDB)               | (wt.%)  | VSMOW)                  | VSMOW)           | (h)    | VSMOW)                          | VSMOW)                     |
| F  | 95.4-98.6   | 176    | -5.52               | 0.10    | -5.87                   | -33.01           | 0.0    | 0.0                             | 0.0                        |
|    |             | 155    | -5.52               | 0.07    | -6.39                   | -34.91           | 19.0   | -0.5                            | -1.9                       |
|    |             | 169    | -5.52               | 0.06    | -6.13                   | -32.99           | 42.5   | -0.3                            | 0.0                        |
|    |             | 111    | -5.52               | 0.15    | -6.40                   | -34.58           | 71.5   | -0.5                            | -1.6                       |
|    |             |        |                     |         |                         |                  |        |                                 |                            |
| G  | 117.0-119.4 | 76     | -5.35               | 0.22    | -5.95                   | -32.67           | 0.0    | 0.00                            | 0.0                        |
|    |             | 90     | -5.35               | 0.13    | -5.96                   | -34.24           | 6.0    | -0.01                           | -1.6                       |
|    |             | 86     | -5.35               | 0.22    | -5.85                   | -33.49           | 22.5   | 0.10                            | -0.8                       |
|    |             | 79     | -5.35               | 0.16    | -6.65                   | -35.98           | 55.0   | -0.70                           | -3.3                       |
|    |             | 92     | -5.35               | 0.16    | -6.01                   | -34.85           | 73.5   | -0.06                           | -2.2                       |
|    |             |        |                     |         |                         |                  |        |                                 |                            |
| Н  | 120.0_123.5 | 151    | -5.38               | 0.08    | -5.93                   | -35.17           | 0.0    | 0.00                            | 0.0                        |
|    |             | 116    | -5.38               | 0.06    | -6.15                   | -34.31           | 27.0   | -0.23                           | 0.9                        |
|    |             | 155    | -5.38               | 0.19    | -6.44                   | -33.93           | 50.5   | -0.51                           | 1.2                        |
|    |             | 119    | -5.38               | 0.11    | -5.79                   | -33.39           | 72.0   | 0.13                            | 1.8                        |
| T  | 140 ( 150 0 | 0.4    | 4.04                | 0.00    | C 41                    | 24.00            | 0.0    | 0.12                            | 0.6                        |
| 1  | 148.6-152.2 | 84     | -4.94               | 0.08    | -6.41                   | -34.98           | 0.0    | -0.12                           | -0.6                       |
|    |             | 113    | -4.94               | 0.14    | -6.18                   | -33.81           | 0.0    | 0.12                            | 0.6                        |
|    |             | 107    | -4.94               | 0.25    | -5.94                   | -33.57           | 6.0    | 0.35                            | 0.8                        |
|    |             | 95     | -4.94               | 0.31    | -5.79                   | -30.86           | 23.0   | 0.51                            | 3.5                        |
|    |             | 71     | -4.94               | 0.15    | -6.77                   | -36.49           | 43.5   | -0.48                           | -2.1                       |
| T  | 177.0.100.5 | 0.2    | 5.20                | 0.16    | 6.44                    | 24.50            | 0.0    | 0.00                            | 0.0                        |
| J  | 1/7.9-180.6 | 92     | -5.30               | 0.16    | -6.44                   | -34.59           | 0.0    | 0.00                            | 0.0                        |
|    |             | 134    | -5.30               | 0.18    | -5.73                   | -32.05           | 52.5   | 0.71                            | 2.5                        |
|    |             | 84.7   | -5.30               | 0.16    | -5.83                   | -32.23           | 77.0   | 0.61                            | 2.4                        |

| 4 | 4 | 0 |
|---|---|---|
| - | - | v |

| Depth
(mm) | Weight
(g) | 238 U
(ppb) | 232 Th
(ppt) | $\delta^{234}$ U (measured a ) | [ 230 Th/ 238 U]
(activity c ) | [ 230 Th/ 232 Th]
(ppm d ) | Age
(uncorrected) | Age
(yr BP c,e ) | $\delta^{234} U_{initial}$
(corrected b ) |
|---------------|---------------|---------------------------|----------------------------|--------------------------------------------------|--------------------------------------------------------------------|----------------------------------------------------------------|----------------------|--------------------------------|---------------------------------------------------------|
| 75.0          | 0.2972        | $164.20{\pm}~0.46$        | 2.2±1.6                    | 61.8± 3.3                                        | 0.06145±0.00047                                                    | $76286{\pm}\ 54629$                                            | $6497{\pm}~55$       | $6429{\pm}~55$                 | 63.0± 3.0                                               |
| 190.0         | 0.2601        | $137.39{\pm}0.26$         | 3.7±1.8                    | 58.1±2.1                                         | 0.06729±0.00041                                                    | $41683{\pm}20343$                                              | 7160±48              | $7092 \pm 48$                  | 59.3±2.1                                                |

Chemical analyses were performed in March 2017. Analytical errors are  $2\sigma$  of the mean.

a  $[^{238}U] = [^{235}U] \times 137.818 \ (\pm 0.65 \ \%)$  (Hiess et al., 2012);  $\delta^{234}U = ([^{234}U/^{238}U]_{activity} - 1) \times 1000.$

 $^{b}\delta^{234}U_{initial}$  corrected was calculated based on  $^{230}$ Th age (T), i.e.,  $\delta^{234}U_{initial} = \delta^{234}U_{measured} \times e^{\lambda 234 \times T}$ , and T is the corrected age.

445  $c[^{230}\text{Th}/^{238}\text{U}]$  activity =  $1 - e^{-\lambda 230 \times T} + (\delta^{234}\text{U}_{\text{measured}} / 1000) [\lambda_{230} / (\lambda_{230} - \lambda_{234})] (1 - e^{-(\lambda 230 - \lambda 234)T})$ , where T is the age. The decay constants are  $9.1705 \times 10^{-6} \text{ yr}^{-1}$  for  $^{230}\text{Th}$ ,  $2.8221 \times 10^{-6} \text{ yr}^{-1}$  for  $^{234}\text{U}$  and  $1.55125 \times 10^{-10} \text{ yr}^{-1}$  for  $^{238}\text{U}$ .

d The degree of detrital 230Th contamination is indicated by the [230Th / 232Th] atomic ratio instead of the activity ratio.

e Age corrections for samples were calculated using an estimated atomic  ${}^{230}$ Th/ ${}^{232}$ Th ratio of 4 ± 2 ppm. These are the values for a material at secular equilibrium, with a crustal  ${}^{232}$ Th/ ${}^{238}$ U of 3.8 with an arbitrary uncertainty of 50%. BP indicates Before Present, where Present

450 is defined as 1950 CE.

Figures

455 Figure 1: Photographs of HSN1 stalagmite.

Half section of the HSN1 stalagmite from Hoshino Cave. The horizontal bar represents 5 cm. The positions of samples (A-J)
 used for the experiments are shown as bold black box. An example of wedge-shaped sub-samples from a layer is illustrated schematically.